# Pharmacogenomics: A Genetic Approach to Drug Development and Therapy

**DOI:** 10.3390/ph17070940

**Published:** 2024-07-13

**Authors:** Rowaid Qahwaji, Ibraheem Ashankyty, Naif S. Sannan, Mohannad S. Hazzazi, Ammar A. Basabrain, Mohammad Mobashir

**Affiliations:** 1Department of Medical Laboratory Sciences, Faculty of Applied Medical Sciences, King Abdulaziz University, Jeddah 22254, Saudi Arabia; rgahwajy@kau.edu.sa (R.Q.); ishankyty@kau.edu.sa (I.A.); mshazzazi@kau.edu.sa (M.S.H.); ammar.basabrain@gmail.com (A.A.B.); 2Hematology Research Unit, King Fahd Medical Research Center, King Abdulaziz University, Jeddah 21589, Saudi Arabia; 3College of Applied Medical Sciences, King Saud bin Abdulaziz University for Health Sciences, Ar Rimayah, Riyadh 14611, Saudi Arabia; sannann@ksau-hs.edu.sa; 4King Abdullah International Medical Research Center, Jeddah 22384, Saudi Arabia; 5Department of Biomedical Laboratory Science, Faculty of Natural Sciences, Norwegian University of Science and Technology, NO-7491 Trondheim, Norway

**Keywords:** pharmacogenomics, genetic approach, drug therapy, drug development, personalized medicine/therapy, human diseases

## Abstract

The majority of the well-known pharmacogenomics research used in the medical sciences contributes to our understanding of medication interactions. It has a significant impact on treatment and drug development. The broad use of pharmacogenomics is required for the progress of therapy. The main focus is on how genes and an intricate gene system affect the body’s reaction to medications. Novel biomarkers that help identify a patient group that is more or less likely to respond to a certain medication have been discovered as a result of recent developments in the field of clinical therapeutics. It aims to improve customized therapy by giving the appropriate drug at the right dose at the right time and making sure that the right prescriptions are issued. A combination of genetic, environmental, and patient variables that impact the pharmacokinetics and/or pharmacodynamics of medications results in interindividual variance in drug response. Drug development, illness susceptibility, and treatment efficacy are all impacted by pharmacogenomics. The purpose of this work is to give a review that might serve as a foundation for the creation of new pharmacogenomics applications, techniques, or strategies.

## 1. Introduction

Pharmacogenetic research over a long period of time has demonstrated how genetic variants affect drug response in a broad way [1,2,3,4,5]. With the increasing number of known functional polymorphisms and the availability of high-density genomic SNP maps, there is hope that pharmacogenetics may be able to optimize patient-specific medications. As genomes and other omics technologies are used more often, the term “pharmacogenomics” has evolved to describe this evolving method of drug discovery and treatment [1,2,3,4,5]. One drug fits all is replaced with “the right drug for the right patient at the right dose and time” in pharmacogenomics, the first step toward personalized medicine. This does not imply that all patients will receive care that is unaffordable. Instead, individuals are categorized into groups according to genetic and other factors that indicate how the disease will develop and how effective a medication will be. When utilizing drug therapy, one must avoid toxicity or an inability to react. A drug obtains a more favorable risk/benefit ratio and has the potential to become the first-choice therapy, increasing its market share, if the incidence of adverse events can be further decreased in the targeted population. Researchers anticipate a growing trend—the cornerstone of personalized medicine—to associate the release of new medications with diagnostic indicators, frequently genetic ones, in order to enhance treatment outcomes for individual patients [6,7,8,9]. Differentiated drug response may be caused by a multitude of variables, such as age, sex, body weight, diet, organ function, infections, drugs, and inheritance. One of the numerous strategies used in personalized medicine is pharmacogenomics, and medical informatics makes it simpler to include pertinent data (Figure 1). Here, we have outlined the key components of pharmacogenomics therapies, which are often simplified processes. Clinical sample collection is the first step, followed by genotyping, analysis, and the discovery of multi-level components linked to complicated human illnesses; in the end are the biological functions or connected pathways. However, because disease processes and pharmacological treatments are complex systems with unpredictable behavior, pharmacogenomics can only make limited predictions. Understanding the vast scope of pharmacogenomics and the obstacles that must be removed in order for tailored medicine to become a reality is therefore difficult.

The implementation of precision medicine may be contingent upon next-generation sequencing methodology and technology [3,10,11]. An opportunistic target capture phase may be employed to decrease the necessary sequencing capacity and improve the coverage of genomic areas of interest. Following sequencing, an appropriate reference, such as the human reference genome or, in the case of tumor biopsies, the patient’s germline genome, is compared to identify the genetic variations present in the sample. Computational techniques or data from previous research can be utilized to predict the functional consequences of modifications detected if they are not fully understood (Figure 1).

As we mentioned the basics of the steps implemented in pharmacogenomics therapeutics in the previous paragraphs, now we introduce the general factors associated with therapeutic failure and drug resistance. A primary factor contributing to patient morbidity and mortality is the variation in pharmacological therapeutic response. Most inpatient and outpatient patients encounter unpleasant medication-related events, such as adverse drug reactions (ADRs) and sub-therapeutic outcomes from pharmacological therapy [3,12,13,14]. Many patient-specific factors, including age, diet, polypharmacy, concomitant diseases, and heritable factors, contribute to these interindividual variances in medication response; a substantial amount of this variability is driven by genetic polymorphisms. The liver, which is the main organ involved in drug metabolism, excretes the majority of drugs. The cytochrome P450 (CYP) class of genes encodes enzymes that perform the majority of phase-I drug metabolism, making them significant drug response modulators. The bioactivation and/or detoxification of the medicine may be significantly impacted by the notable variation of CYP genes both within and across populations [2,3,15].

Thus, after presenting the basic steps and the factors associated with pharmacogenomics therapeutics, we feel that the most critical step in this direction is to predict the potential and most appropriate biomarkers. Some hypotheses suggest that pharmacogenomic biomarkers that might predict drug response could be very useful for enhancing molecular diagnostics in ordinary clinical treatment. It is crucial to distinguish between somatic cancer genome biomarkers, which affect how cancer cells respond to medications, and germline biomarkers, which affect the pharmacokinetics and pharmacodynamics of systemic pharmaceuticals. Drug response variations have been related to epigenetic changes in DNA or histones in addition to hereditary variables. In oncology, the overexpression of the drug efflux transporter has been connected to epigenetic modifications in cancer cells that underpin treatment resistance. Blood may include DNA that has undergone epigenetic alteration, offering a unique means of tracking the development of drug resistance and the effectiveness of therapy [3,10,16,17,18,19,20,21,22,23,24,25,26,27,28,29,30,31,32,33,34]. Another use for it is the classification of tumors. Furthermore, pharmacological modulators of the epigenetic machinery have been effectively applied to the treatment of cancer, mostly as adjuvants to increase tumor sensitivity to chemotherapy administered as routine care. We give a comprehensive update on this topic by reviewing current studies. An overview of the genetic markers that forecast medication response and direct therapeutic decision-making, such as medication choice and dose, is provided in this article. We also talk about recent technological developments that make it easier to find and use biomarkers [3].

## 2. Pharmacogenomics

Pharmacogenomics is one tool that the pharmaceutical industry may use. It represents a major advancement in medical history. Finding novel targets for new drugs, improving efficacy and reducing adverse drug reactions, correlating genotype with clinical genotype, and pharmacogenetically profiling individuals to forecast drug response and sickness risk are its main objectives. Most drugs used to be developed with the population in mind, rather than being particularly formulated for each patient. By countering that trend, pharmacogenomics aims to improve the effectiveness and safety of drugs while concentrating on therapy. Instead of focusing on the outward manifestation of the disease, or the phenotype (the signs and symptoms), pharmacogenomic treatment examines and treats the genotype. Pharmacogenomic research will eventually be included into drug discovery and development, resulting in a considerable reduction in the cost of medicine development [1,35,36,37,38,39,40]. Additionally, it will ensure the safety of the clinical study and reduce the number of failures. Consequently, many potential drugs that would be lost due to the effects on the outliers in a research study can be kept when the pharmacogenomic study is used in the future (Figure 1).

Treating each patient as an individual and forecasting the clinical result of various therapies for various patients are the two main objectives of personalized medicine. Pharmacogenomics is a fundamental component of personalized medicine. The fundamental idea is that a patient’s characteristics, such as age, gender, and/or concurrent medicines, as well as environmental variables, genetics, and epigenomics, all have an impact on the interindividual variability in drug response [41]. Advances in pharmacogenomics, often known as “omics” methods, have transformed our understanding of the genesis and susceptibility of diseases and have great promise for the development of new treatment approaches. Ivacaftor is only approved for use in the treatment of cystic fibrosis in individuals who have the particular G551D genetic mutation in their CFTR gene. Encoded by the CFTR gene, a protein that controls the body’s transportation of water and chloride breaks down in cystic fibrosis. Ivacaftor works by activating the CFTR protein, which improves lung function [42].

Targeted treatment is still a relatively new notion, even though examples like this one and others (vemurafenib, which inhibits the BRAF V600E mutation in malignant melanoma) imply that the blockbuster paradigm of drug development is ending. One explanation for this is that monogenic pharmacogenetic traits usually do not account for variations in a complex phenotype such as drug response. In addition to having several off-target effects, most medications used today have multiple targets, according to data from drug-target network research [9,10]. Gaining a grasp of genome-wide techniques like metabolomics, epigenomic profiling, and sequencing is essential to comprehending the molecular architecture of disease genesis and/or therapy response. Although genome-wide association studies (GWAS) have revealed several new biological pathways, this method has drawbacks in that the majority of alterations linked to clinical phenotypes—like adverse pharmaceutical reactions—are often not causative. One may reasonably expect that pharmacogenomic research would benefit from the combination of several omics technologies [9,10,30,43,44,45]. Recently, multi-omics research has proven useful in locating potential novel targets for therapy (Figure 1).

Numerous significant pharmacogenomics applications have been licensed by the FDA and are already being utilized in clinical practice. These applications include warfarin and CYP2C9/VKORC1, cetuximab/panitumumab and KRAS, vemurafenib and BRAF, abacavir and HLA-B*5701, carbamazepin and HLA-B*1502, and thiopurines and TPMT. To assess the usefulness of alternative options (like tamoxifen) in clinical settings, more research is needed. To better translate pharmacogenomics from lab to bedside, a more thorough examination of the dynamic relationship that may exist between a patient’s genome and their phenotype (e.g., pharmaceutical response), which may change over time, is also required (Figure 1). Recent research using warfarin algorithms has shown that the addition of non-genetic variables, such as environmental and clinical covariates, can provide a substantial amount of additional phenotypic data to enhance the precision of a treatment decision. Age, sex, body mass index, diet, genetic variation in CYP2C9 and VKORC1, concurrent drug therapy, ethnic background, and food all affect how much warfarin is needed [5].

In the past several years, pharmacogenomics has drawn a lot of interest, and functional genomic techniques will likely prove to be an invaluable resource for predicting clinical outcomes in the future. Multi-omics technologies have lately sparked interest in the field of pharmacogenomics research, which is evidently quite fascinating. However, a concentrated effort is required to link the knowledge of basic and clinical researchers with other sectors, including the healthcare community, regulators, and commercial partners, in order to demonstrate the therapeutic benefit of pharmacogenomics in the majority of medical specialties [2,3,4,5].

## 3. Genetic Causes of Individual Variability in Drug Response

The wide range of pharmacological response and toxicity as well as phenotypic variability prevent a medication from being used in clinical settings (Figure 2). Less than 70% of patients receive a satisfactory response with some of the most advanced drugs available today, and a significant portion endure adverse effects. For many patients, this leads to a poor risk/benefit ratio. Understanding variability requires an understanding of pharmacokinetics (PK) and pharmacodynamics (PD), two domains that provide quantitative assessments of drug exposure and impact. PD is primarily focused on drug targets (receptors and enzymes), downstream signaling pathways, and pharmacological response, whereas PK is more concerned with absorption, distribution, metabolism, and excretion (ADME). There are several polymorphism genes that are crucial for PK-PD [46,47,48]. Because ADME regulates medication exposure, medication level monitoring generates phenotypic indications that are useful for tailored therapy [49]. High-throughput technologies were previously used in PK screening to find predictive biomarkers of toxicity or efficacy in cancer therapy. If these biomarkers were used in clinical practice, they might lead to the development of individualized therapies based on a patient’s genetic composition. The application of pharmacogenomic technologies and the practical efficacy of pharmacogenetic screening might improve patient safety by identifying biomarkers related to drug metabolism for customized treatment. Pharmacogenetic pediatric research has shown encouraging findings, despite the fact that pharmacogenomic studies were conducted in adult cohorts. According to a meta-analysis, polymorphic drug-metabolizing enzymes have been linked to serious negative pharmacological consequences [17,50,51]. Protein treatments encompass a wide range of biologics, such as therapeutic replacement enzymes, fusion proteins, and antibodies. Since their conception, they have transformed the way that a variety of illnesses, including cancer, autoimmune, inflammatory, respiratory, vascular, and neurological disorders, are treated. Protein therapies are frequently the subject of in vivo pharmacokinetic, pharmacodynamic, and effectiveness research; however, studies that pinpoint the critical variables influencing the absorption, distribution, metabolism, and excretion (ADME) features of these agents have not received enough attention. The extensive characterization and comprehensive investigation of their ADME characteristics are essential to assist drug research and development procedures for the creation of safer and more potent biotherapeutics. This suggests a potential tactic to reduce the probability of unfavorable outcomes when utilizing genetic data [4,10,24,52].

Drug transporters, which are extensively engaged in ADME and drug targeting, are encoded by several hundred genes. Though little is known about their effects, a number of functional polymorphisms appear to alter pharmaceutical response. It is exceedingly difficult to analyze the effects of polymorphisms in the genes that encode drug receptors. Activating mutations could be an exception, particularly if they impact tyrosine kinases, which are crucial for the growth of cancer. For example, responsiveness to gefitinib is generally correlated with activating mutations of EGFR, but the constitutive activation of the fusion protein BCR/ABL (resulting from chromosomal translocation in leukemia) confers a notable sensitivity to imatinib. ErbB2 over-expression is necessary for the efficacy of herceptin therapy in the treatment of breast cancer.

## 4. Future of Genotypes in Drug Therapeutics

If there is a strong and frequent genetic component, prospective genotyping may be advised if obtaining the best available pharmacological therapy might have disastrous consequences. Finding the genetic factors causing varying drug responses may be enough in many cases to lower the likelihood of serious side effects. The therapeutic hypothesis is supported by evidence from human genetics, which raises the probability that a medicine will be successful in clinical trials. Numerous alleles with a variety of impact sizes are produced by common and rare disease genetics, and these alleles can be used as proxies for a drug’s effect in a given condition. A wealth of genetic data about humans has been made available recently through large-scale population collections and whole genome sequencing techniques, supporting the choice of therapeutic targets. These methods will have a greater impact on several phases of a drug development program as the variety of phenotypes profiled broadens and additional alleles from people throughout the world are found. Because genotyping in a therapeutic facility presents practical problems in addition to legal and financial concerns, several opinions have been voiced regarding prospective genotyping in this case. Alternatively, the judicious monitoring of the white blood cell count may be adequate to avoid major toxicity. It is clear that there are ethical, legal, economical, and medical issues to take into account when using potential genotyping at the bedside [53,54,55,56].

## 5. Drug Response

A complex phenotype known as clinical drug response results from the interaction of several variables, including genetic, clinical, environmental, and demographic ones (Figure 2). Due to this intricacy, there is a significant interindividual heterogeneity medication response, which can have an adverse effect on both effectiveness and toxicity and lead to a wasteful use of the scarce healthcare resources. Through genotype-informed prescription and monitoring guidelines, pharmacogenomics—the research and practical use of the genetic drivers of drug response variation—aims to maximize therapeutic effectiveness and decrease adverse drug responses. A number of approved cardiovascular medications, such as simvastatin (SLCO1B1), warfarin (VKORC1, CYP2C9, CYP4F2), and clopidogrel (CYP2C19), have documented pharmacogenomic relationships [57,58].

Two definitions of interindividual variability in drug response are the need for a range of doses to achieve an effect of a defined intensity in each patient or the occurrence of an effect of varied strength in different people receiving a specific medication dosage. Drug reaction is classified into four categories: toxic impact, no effect or therapeutic failure, unfavorable effect, and intended good effect (efficacy). The dosage of the medication has a special bearing on side effects and therapeutic failure. For precision oncology, medication response prediction based on cancer genetic profiles is crucial. The majority of medication response prediction models now in use were constructed using immortalized cancer cell line drug screening data, which often have different genetic profiles from patient malignancies. Patient-derived organoids, or PDOs, are becoming more and more popular as a platform for accurately simulating patient cancers [2,6,53,57,59].

The most effective method for treating complex diseases like cancer and HIV/AIDS is no longer thought to be single-agent treatment that targets a single receptor. But when many medications are used concurrently, there is a higher chance of drug–drug interactions, which might lead to unexpected and difficult-to-identify adverse effects [9]. For instance, when CYP2D6 poor metabolizers take medication A with another prescription that inhibits CYP2C9, the drug’s metabolism—which is metabolized by both CYP2D6 and CYP2C9—is significantly decreased. Ritonavir is used as an antiviral “boosting” medication when up to three antiviral medications are administered concurrently for anti-HIV therapy. Ritonavir is a strong inhibitor of membrane transporters including CYP3A4 and Pgp (MDR1), depending on the mechanism. This allows other antivirals that are also carried by Pgp and processed by CYP3A4 to be taken at lower dosages; however, the dosing becomes unpredictable. Furthermore, to reduce the lipodystrophic side effects of the antivirals, the majority of patients are co-medicated with statins, antidepressants, and antibiotics [53]. Serious side effects result in a high frequency and intensity, which are probably influenced by polymorphisms in genes linked to ADME. “One gene, one drug” strategy may make it challenging to demonstrate a causal relationship since effects are distributed over a network of interactions. Instead, integrating overall harmful effects with functional variations in several genes requires a systems approach. We propose a medical informatics strategy that evaluates all side effects, particularly those involving sizable patient populations, in relation to the most common pharmacogenetic markers.

Combinatorial treatment has been used in the past to diagnose various tumors (e.g., advanced non-small-cell lung cancer: nivolumab plus ipilimumab). In this case, nivolumab with ipilimumab had a higher response rate than nivolumab monotherapy, especially for patients whose tumors expressed programmed death ligand 1 (PD-L1) [60,61,62].

## 6. Genetic Causes Associated with Phenotype Variations

The stability and processing of mRNA, the structure and function of proteins, and the regulation of gene expression can all be impacted by modifications in the DNA sequence. Extensive studies on genetic variation indicate that polymorphisms affecting cis-regulatory genes are far more common than those altering key protein structure and function. Even if the bulk of them are yet unknown, almost all genes are predicted to contain polymorphism(s) at one or more cis-regulatory sites, which can be found anywhere in the extended gene locus. During mRNA processing, genetic variants also impact alternative splicing and mRNA stability. According to current estimations, 35–59% of human genes experience alternative splicing. Few polymorphisms, such as a synonymous SNP in the dopamine DRD2 receptor, have been demonstrated to alter mRNA stability, despite the fact that many polymorphisms, such as mutations in CYP2D6, have already been found to affect splicing. Nevertheless, most SNPs may impact mRNA folding and, therefore, mRNA stability, processing, or translation, according to computational studies of mRNA folding [4,9,18,31].

Numerous phenotypic variations may be explained by cis-acting polymorphisms that impact mRNA functions, according to previous research and assessments. Typically, this results in an imbalance between the expression of one allele and the other in the production of mRNA (allelic expression). To find an imbalance in allele expression, a method that uses the PCR amplification of genomic DNA and mRNA (as cDNA) of a transcribed region of the gene containing a common marker SNP can be applied. The next step is to determine the allelic ratios in both DNA and mRNA. Each allele has an own control system that eliminates transacting effects. A large number of marker SNPs might be used to see if splicing events change polymorphically. Since the target tissues have different controls on transcriptional and mRNA processing, the test must be carried out there. In cases where trans-acting mechanisms (transcription factors) cause genetic variability in mRNA levels, identifying the underlying cis-acting polymorphisms upstream in the signaling cascades is essential. Allelic expression imbalance analysis’s low repeatability may be the reason for its limited application, despite the fact that it has the potential to be a useful tool for discovering cis-acting factors. Many important genes can have different expression patterns, which might result in a disease or influence the course of treatment. It has been demonstrated that epigenetic changes can also cause an imbalance in allelic mRNA. One way to identify interindividual differences in mRNA processing and gene expression is to evaluate the allele DNA to mRNA ratio. This approach yields quantitative characteristics that may be utilized to identify the cis-acting variables and is sensitive to each of these processes [8,18,36].

## 7. Knowledge Gap about the Genetic Contribution to Phenotype Variations

Even for genes that have been well studied, the entire genetic variability remains unknown. A functional polymorphism is frequently employed in clinical studies after receiving experimental validation, but its relative contribution to overall genetic variability is never assessed. Most genes have a large number of functional polymorphisms [57,63,64,65,66]. For example, various mood and cognitive impairments have been linked to the serotonin transporter gene, SERT. By using a reporter gene test and performing in-depth analysis in association studies, it has been demonstrated that a difference in the promoter region influences the levels of SERT mRNA in lymphocytes. Inconsistent data, however, suggest that the LPR genotypes in the central nervous system have varying degrees of SERT expression. SERT is mostly expressed in neurons found in the pontine region of the brain stem. While non-synonymous mutations in the SERT coding region are uncommon, other regulatory polymorphisms might influence the likelihood of developing a disease or how well a therapy works. A quantitative evaluation of the penetrance of SERT polymorphisms is required when treating mental illnesses or using certain serotonin reuptake inhibitors, which are often prescribed antidepressants with a high rate of patient satisfaction. Although haplotypes can be a helpful tool in combining the impacts of many functional SNPs in phase, like epistasis, they might not provide all the genetic information in different patient groups. Finding any functional polymorphism that manifests often enough in the target group must be our main objective [18,31,42,53,67].

## 8. Epigenetic Effects and Regulation of Gene Expression at the mRNA and Protein Level

Animal cells come in a variety of forms and are the basic components of all multicellular organisms. Although classification techniques remain ambiguous, substantial advances have been made in the characterization of cell types. We provide an evolutionary description of a cell type so that it may be distinguished and compared within and across species. The transcription factors’ “core regulatory complex” (CoRC) has developed in ways that allow it to recognize newly developing sister cell types, support their independent development, and regulate apomeres, or characteristics exclusive to a certain cell type. These alterations are essential for identifying the cell type. We discuss the distinctions between developmental and evolutionary lineages and provide a future research agenda. Even in the absence of genomic DNA polymorphisms, altered gene expression can be passed down from generation to generation through chromatin remodeling or imprinting, as well as during somatic cell divisions. Here, we incorporate a figure from the previous work [1] which presents the simple regulatory mechanism for cell-type identity. Here, we can see how different levels of regulation take place (Figure 3). The methylation of CpG islands and modifications to histones through acetylation and methylation are the main processes behind these transmissible characteristics. Global methylation is also necessary for X chromosome inactivation, which is regulated by the Xist transcript and accounts for variations in gene dosage between males and females. Allele expression is uneven as a result of the frequent skewness in X-inactivation. According to recent research, epigenetic alteration affects disease broadly and may potentially have therapeutic benefits [68,69]. Extended manic and depressive phases of bipolar disorder may be brought on by metastable, reversible epigenetic changes to gene regulation. Histone acetylation is increased and CpG methylations are reversed by decitabine and HDAC inhibitors in an attempt to force the expression of suppressor genes. This is because the same epigenetic processes that inhibit tumor suppressor genes also seem to have an effect on cancer. On the other side, the response to anticancer treatment with cisplatin and BCNU is improved by the methylation of the MGMT promoter, an enzyme that repairs DNA. While it is evident that epigenetic changes have an impact on illness and treatment results, there is currently not enough information to use this understanding in a tailored medical setting in the future [2,7,70].

The astounding complexity of gene regulation and translation has been shown by recent studies on small regulatory RNAs, including antisense transcripts from the opposite DNA strand of many genes, siRNA mechanisms, and the emerging science of microRNAs. With up to 1000 microRNAs in the human genome, each of which targets many genes, one may anticipate that microRNAs have a significant role in both disease and therapy outcomes. Specifically, microRNAs may be involved in chemosensitivity or resistance brought on by chemotherapy. Subsequent investigations will explore the function of short regulatory RNAs in the progression of illness and the efficacy of treatment [67,71].

## 9. Summary of Computational Approach in Pharmacogenomics and Drug Development and Therapeutics

Drug delivery schedules have a major impact on how well cancer therapies work; mathematical models of population dynamics and treatment responses may be used to provide mechanistic insights and optimum drug administration regimens. However, a major challenge is the appropriate interpretation and bioinformatic processing of increasingly complex multi-omics data sets. The operation of biological networks is greatly affected by mutations in the coding sequence or expression of genes, as well as transient responses to external signals at the level of protein activity, posttranslational modification, stochastic processes, etc. It is believed that using genomics by itself is insufficient for research and drug development. Thus, several one-dimensional biomolecular-omics data sets and patient history may be connected utilizing an integrated systems pharmacy approach to improve our comprehension of the biology underlying illnesses and drug-response phenotypes. In the end, this kind of strategy ought to lead to the discovery of new therapeutic targets [23,29,72,73,74,75,76]. We also outline the fundamental tools, techniques, and software in Table 1 and Table 2. Moreover, the multi-omics data integration is summarized in Figure 4 [31]. Ritchie M.D. et al. have presented highly relevant work related to methods for multi-omics data integration which could be most appropriate for pharmacogenomics and personalized therapeutics.

To achieve the full potential of this approach, the cancer community has to get over the challenges of implementing this type of work in clinics. Approximately half of the patients do not react as expected to pharmaceutical therapy. Heritable variables account for a significant portion of these interindividual variances, and there is an increasing number of connections between genetic polymorphisms and pharmaceutical response patterns. Significantly, the pharmacogenes’ genetic landscape is incredibly complicated, with tens of thousands of uncommon genetic variations. This has been revealed by the recent, rapid breakthroughs in next-generation sequencing technology. Given the high frequency of these uncommon variations observed in each individual, it is expected that they play a major role in the genetically encoded interindividual variability in the effects of pharmaceuticals. Since the problem is now so big that a complete experimental characterization of these variations is no longer possible, the primary challenge is to comprehend the functional significance of variants. An outline of the key ideas and advancements in the creation of computational prediction techniques for figuring out how changes in amino acid sequence impact the transporters and enzymes involved in drug metabolism is given here. As is now widely known, recent discoveries regarding the functional implications of non-coding changes, such as those to splice sites, regulatory regions, and miRNA binding sites, seem valuable for the development of pharmacogenomics- and genetics-based medicines. We believe that within a precision medicine framework, the multidisciplinary approach will offer a helpful toolset to enable the inclusion of a wide variety of unusual genetic variability in drug response predictions.

For this reason, computer prediction algorithms are often employed to estimate the functional impact of genetic variations when feasible experimental procedures are not available. The majority of these algorithms seek to forecast how modifications resulting in amino acid substitutions would affect function. However, in recent times, there has been a noticeable progress in the understanding of non-coding mutations that impact splice sites, enhancers, promoters, or miRNA binding sites. a list of the characteristics that are currently measurable by computer prediction algorithms. Whether genetic alterations are found in the coding sequences of the gene, in untranslated sections, in putatively regulatory sequences, or inside introns determines the significance of many characteristics and attributes such as RNA binding protein, non-sense-mediated decay, intronic splicing enhancer/silencer, and exonic splicing enhancer/silencer [67,71].

Most prediction tools base their judgments, at least partially, on the evolutionarily conserved sequence in issue; prediction algorithms are often trained on collections of harmful variants. Most notably, though, pharmacogenes are unique in that they have limited evolutionary conservation and are typically unrelated to human illness. These variances confound the understanding of pharmacogenetic variations. With an emphasis on their applicability for pharmacogenetic predictions, we also reviewed computational methods for the functional interpretation of genetic variations in this instance. We came to the conclusion that one of the most significant areas for the therapeutic use of NGS-based genotyping is still the development of computational tools, which are crucial for the functional interpretation of an individual’s pharmaco-genotype [18,19,48,68,77,78,79,80,81,82,83,84,85,86]. Finally, we present a summary for pharmacogenomics-based therapeutic studies (Figure 4).

**Table 1 pharmaceuticals-17-00940-t001:** Essential fundamental methods [87] for pharmacogenetics and genomics genotype analysis [2,3,4,17,48,55,58,63,87,88].

Method	Short Description and Purpose
Sanger dideoxy (end terminal) sequencing	Analyzing DNA sequences and finding novel polymorphisms.
Denaturing high performance liquid chromatography (DHPLC)	Ion-pair reverse-phase HPLC can be used to differentiate the differentially shaped hybrid molecules (homoduplex versus heteroduplex) that result from the combination of variant and wild-type DNA in order to detect polymorphisms.
PCR-RFLP	Restriction endonucleases, which are enzymes unique to a certain sequence, cut the amplified polymorphic genomic area. The resultant fragments are indicative of the genotypes and are subjected to electrophoresis analysis.
Pyrosequencing [89,90]	A DNA sequencing technique that makes use of the sequencing by synthesis concept. It is used in DNA methylation studies and SNP genotyping. The “next generation” of large-scale DNA sequencing, which can sequence more than 100 million base pairs a day, is based on the same premise as this approach.
Single-base (primer) extension (also known as mini-sequencing) [91]	The 3’ end of short oligonucleotides is annealed directly upstream of the polymorphism site. A combination of (fluorescently labeled) ddNTPs without dNTPs is used to elongate a single base alone. The MALDI-TOF detection technique or sequencing can be used to identify the products. It is used as a multiplex reaction for genotyping SNPs.
DNA microarrays [92]	Using microarray solid-phase attached DNA molecules, a single sample may be genotyped for many SNPs—up to a million—at once. This method is utilized in research on genome-wide associations.
RNA/cDNA microarrays [93]	Utilized to measure the quantity of transcripts in a single sample or to compare two samples while performing gene expression analysis. Beneficial for quantifying a large range of distinct transcripts in a single sample, including those found throughout the genome.
PCR [93]	PCR is a fundamental method used in nearly all modern genomic and pharmacogenetic analyses.
qPCR [94]	Employing different fluorescence quenching or fluorescence energy transfer techniques to detect the development of the PCR product while the PCR reaction is ongoing in order to genotype individual SNPs in a variety of samples.
qRT-PCR [94]	Used following a reverse transcription procedure to measure the number of transcripts in a sample. Helpful for quantifying RNAs in large quantities of samples.

**Table 2 pharmaceuticals-17-00940-t002:** Bioinformatics databases and software tools [87] for pharmacogenetics and genomics [2,3,4,5,9,17,46,48,49,55,58,63,88,95].

Aim	Computer Solution	Website [Accessed on 3 July 3024]
Databases		
Human genome [87]	National Center for Biotechnology Information in the USA (NCBI)	www.ncbi.nlm.nih.gov/genome/guide/human/
	Ensembl	www.ensembl.org/Homo_sapiens/
SNP databases [96,97,98,99,100]	dbSNP at NCBI	www.ncbi.nlm.nih.gov/snp/
	Japan database JSNP	https://dbarchive.biosciencedbc.jp/data/jsnp/LATEST/README_e.html
Pairwise linkage disequilibrium and haplotypes	HapMap project [101]	www.hapmap.org
Gene expression analysis	Gene Expression Omnibus (GEO) by NCBI [102,103]	www.ncbi.nlm.nih.gov/geo/
Metabolic pathways	Kyoto Encyclopedia of Genes and Genomes (KEGG) [104]	www.genome.jp/kegg/
Software		
Homology search	BLAST at NCBI [105]	www.ncbi.nlm.nih.gov/BLAST/
Sequence alignment and identification of new SNPs	Gap5 (part of Staden package) [106]	http://staden.sourceforge.net/
Haplotype mapping (phasing)	Phase, Fastphase [107,108]	http://stephenslab.uchicago.edu/software.html (there is also a new program for imputation of analyzed to in silico linked SNPs)
Pairwise linkage disequilibrium and visualization of Haplotype blocks	Haploview [109,110]	www.broad.mit.edu/mpg/haploview/
Extended haplotype homozygosity (EHH)	Sweep [111]	www.broad.mit.edu/mpg/sweep/
Analysis of SNPs affecting promoter function	TRANSFAC [112,113]	https://bioinformatics.umg.eu/
Analysis of SNPs affecting splice sites and ESEs	Automated Splice Site Analyses (Children’s Mercy Hospitals Missouri, USA) [114]	http://isplice.cmu.edu.tw/index.htm
	ESEfinder 3.0 (Cold Spring Harbor Laboratory) [115]	http://rulai.cshl.edu/cgi-bin/tools/ESE3/esefinder.cgi?process=home

### 9.1. Sequence Analysis, Predictions, and Functional Impact of Variants

The degree of conservation is a measure of how important a sequence is for the structure and function of the associated gene product. It is computed by examining the evolutionary variation dynamics of DNA or amino acid sequences among homologs. Therefore, regions with high evolutionary rates are thought to be crucial, whereas slowly evolving, or conserved, sequences show selection pressure against variation in these areas and, consequently, unfavorable consequences in the event of a mutation. Evolutionary conservation is a parameter used by most computational prediction systems to distinguish between harmful and benign variants. While some algorithms focus on nucleotide sequence alignments or a combination of the two, most systems that focus on functional interpretation of missense changes employ alignments of amino acid sequences. Amino acid sequence alignment has been demonstrated to be effective in missense variant analysis; however, genomic sequence alignments provide more flexibility and allow functional interpretations to be extended to variant classes, such as synonymous and regulatory variants, that do not alter the amino acid sequence. Notably, commonly employed conservation-based function-dependability predictions ignore sequence interdependencies. Nonetheless, it has been shown lately that predictive accuracy is enhanced by the explicit integration of residue dependency information from various sequence alignments, underscoring the advantage of merging variation interaction data with conservation-based functionality predictions [9,34,116,117,118,119,120,121,122,123].

miRNAs have a major role in the regulation of mRNA stability and translation. Ten percent or more of all SNPs are located at conserved miRNA binding sites in 3′-UTRs, which promote miRNA-mRNA interaction and may influence complementary miRNA-mRNA pairing. Moreover, it has been shown that miRNAs significantly alter the gene expression patterns of ADME. Consequently, one of the most important factors in deciding the destiny of the linked transcript is the functional interpretation of genetic changes inside miRNA target sites [124,125,126,127,128,129,130]. Many databases, including the polymiRTS Database 3.0 and MirSNP, offer helpful resources that can be used to evaluate the potential significance of genetic polymorphisms in UTRs. These databases include a collection of experimentally confirmed SNPs and indels in both the miRNA target sites and the miRNA seed regions responsible for mRNA binding. Additionally, a variety of additional public SNP impact prediction algorithms are accessible [8,18,58,131].

Several computational methods may be used to forecast the potential disruption of the miRNA-mRNA pairing for a certain variation in the absence of experimental evidence. MicroSNiPer and ImiRP use vast variation databases to compare the mutant 3′-UTR sequences with one another in order to find and anticipate such disruptions. In a similar vein, mrSNP has the ability to forecast the impact of any mutation found in NGS-based studies on the interaction between target transcripts and miRNA. It is noteworthy that a significant fraction of predicted miRNA targets seem to be false-positive, indicating that similar issues can potentially arise for research utilizing miRNA-target databases lacking strong experimental validations. Inverse techniques, which estimate the impact of genetic variations in suspected miRNA target sites and search for potential negative effects in changes in miRNAs or pre-miRNAs, are facilitated by a number of web-based applications. The reader is directed to current reviews and internet resources for a more thorough collection of variant interpretation tools connected to miRNA. The range of cutting-edge techniques that go beyond the prediction of miRNA binding sites now includes the impact of UTR variations on the binding of RNA-binding proteins (RBPs), translational efficiency, and ribosomal loading.

### 9.2. Analysis of Regulatory Variants

Considering non-coding areas significantly expands the analytic space accessible for computer predictions, since they comprise more than 99% of the human genome. Variants in non-coding regions may alter the local chromatin structure or the transcription factor binding affinity of regulatory elements, including enhancers, silencers, insulators, and promoters. Accurately predicting the functional impact of such changes is a major challenge in human genetics [9,67,76,132,133,134].

Many approaches have been put up to interpret noncoding variances. The first approaches, such as GERP++, SiPhy, PhyloP, PhastCons, or SCONE, used sequence alignments to restrict evolution. Conservation of regulatory areas can only be a poor predictor of the functional impact of SNVs in regulatory regions, as was shown when no extra constraints were found in regulatory elements at the level of DNA sequence despite conserved transcription factor binding. Therefore, to increase prediction quality, functional genomics parameters such as transcription factor binding profiles, DNase I hypersensitive sites, information about histone modifications, sequence, and genic context were added to conservation metrics. These massive data sets were subjected to a range of machine learning techniques, including GWAVA, CADD, FATHMM, DANN, DIVAN, and Genomiser, to produce a number of ensemble classifiers with the goal of differentiating between pathogenic and neutral variations [3,4,5].

### 9.3. Overall Functional Relevance and Impact

Thanks to technological developments, NGS is now often employed in clinical diagnostics and medical genetics. However, the practical use of NGS-based pharmacogenomics is still far behind. Most importantly, in order to fully capitalize on the main advantage of NGS-based genotyping, namely, the identification of the entire spectrum of the individual’s genetic portfolio, tools that facilitate the translation of these variability data into functional implications and clinical recommendations would need to be in place. Pharmacogenomic phenotypes are usually more difficult to detect because they are context-specific, such as exposure to particular medications. In contrast, the presence of distinct phenotypic alterations in the affected patient and the ability to perform comparative genomic analyses of unaffected family members aid in the identification of rare putatively deleterious mutations in congenital diseases. Furthermore, reliable computational prediction methods are desperately needed to bridge this gap due to the lack of experimental characterizations or drug response associations that facilitate the functional interpretation of rare variants.

The functional implications of missense mutations that are relevant to function have been studied the most. The predictions of the associated techniques are based on evolutionary conservation and the structural features of the polypeptide that each gene encodes. Importantly, evolutionary conservation is not a good signal of the effects of variations in genes with little selective pressure, like the majority of pharmacogenes, even though it is a valuable tool for determining the harmfulness of a variant or its impact on organismal fitness. Computer predictors may be trained using ADME missense variants once conceptual issues have been identified. Moreover, many methodologies have been established to examine the functional implications of mutations in non-coding genomic areas, which are progressively acknowledged as a primary contributor to interindividual variability. An increasing number of algorithms are now accounting for a wide range of characteristics, including splicing modulation, effects on transcriptional processes, the disruption of transcription factor binding sites or polymerase loading, and effects on translational efficiency or miRNA binding. The majority of these algorithms have not been independently benchmarked, but rather trained on sets of pathogenic variants, even though these advancements offer a methodological toolkit to thoroughly describe each class of genetic variant. Therefore, more research is needed to determine their ability to predict outcomes for pharmacogenetic analysis [9].

The ability to predict drug metabolism characteristics based on a person’s genotype has come a long way over the past decades. Traditional methods evaluate drug response by utilizing data from a small number of candidate variations for which thorough in vitro or in vivo characterization data are available [2,31,53,57,135]. The functional consequences of a wide range of uncommon genetic variations have not been investigated, despite the fact that this method has been successful in incorporating common pharmacogenetic variability into clinical decision-making. Utilizing Whole Exome Sequencing (WES) to thoroughly examine the genetic landscape of pharmacogenomic sites and also incorporate uncommon variations, extremely complex investigations have begun. As previously mentioned, the study was limited to pharmacogenetic missense variations, and the effects of SNVs with unclear functional relevance were evaluated using computer models trained on pathogenic data sets. Consequently, these approaches have a rather low predictive ability, even if they represent a substantial development in the further customization of genotype-guided therapy decisions.

## 10. Conclusions and Future Perspectives

The pharmaceutical sector could find pharmacogenomics to be a helpful tool. It is a significant advancement in the history of medicine. Finding new targets for innovative medications, reducing adverse drug responses, improving effectiveness, and using pharmacogenetic patient profiles to forecast illness risk and treatment response are some of its primary goals. In the past, the whole public was considered while developing most medications, not specific patients. Pharmacogenomics helps to focus on therapy, improves pharmaceutical efficacy, and reduces adverse effects by opposing this tendency. Pharmacogenomic treatment looks at the genotype and addresses it, as opposed to focusing on the disease’s external expression, or what doctors call the phenotype. In the end, pharmacogenomic research will be incorporated into the procedure to lower the expense of medication development. It will also lower the number of failures and guarantee the safety of the clinical investigation. Therefore, when pharmacogenomic study is employed in the future, many promising medications that would be lost owing to the impacts on the outliers in a research study can be maintained.

Although practical use is still several years off, the field of pharmacogenomics is making progress in understanding medication response. Before pharmacogenomics can be efficiently used in pharmacological therapy, there are a number of challenges to be addressed. The manner in which medications and drug combinations interact with the body is determined by several routes. Pharmacological interactions may have unanticipated effects related to genes with polymorphisms. A systems analysis of medial informatics to integrate all pertinent data is necessary for the genetic study of the entire pharmacological response. It is essential to have a quantitative understanding of how the genetic factors contribute to the target phenotype. In addition to the molecular genetic analysis of polymorphisms affecting the main structure of proteins, we propose the systematic use of allelic expression imbalance for the quantitative assessment of cis-acting factors in transcription and mRNA processing.

We must assess how small regulatory RNAs and epigenetic factors contribute to interindividual variability. A regulatory framework is necessary to guarantee that pharmacogenomic data are integrated into the development of medications and post-approval surveillance. Because the implications of genetic and genomic data are still poorly understood, the FDA created a “safe haven policy” to encourage pharmaceutical companies to use genomic data for the New Drug Approval process without fear of delays or other regulatory measures. These types of data will be more important in the medicine approval process as research advances. Pharmacogenetic data on pharmaceutical package inserts have made genetic information more accessible to physicians and patients. Finally, it should be noted that pharmacogenomics is an increasingly useful technique for understanding interindividual heterogeneity in drug response and toxicity. However, significant advancements in pharmaceutical therapy necessitate an integrated systems approach that enhances customized care through the use of medical informatics.

## Figures and Tables

**Figure 1 pharmaceuticals-17-00940-f001:**
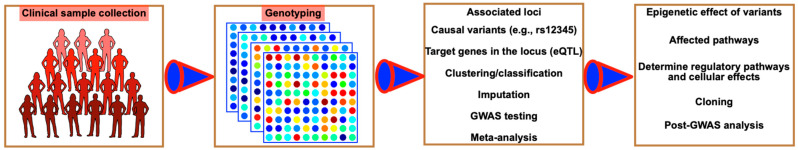
Summary of the steps taken in pharmacogenomics therapeutics. It starts from the integration of multi-omics data (the generation and analysis of large data sets by different high-throughput approaches) and proceeds through pathway-level understanding, pathway–pathway interactions (pathway crosstalk), and network-level understanding, unraveling the integrated mechanisms and predicting the optimal putative biomarkers in the case of cancer.

**Figure 2 pharmaceuticals-17-00940-f002:**
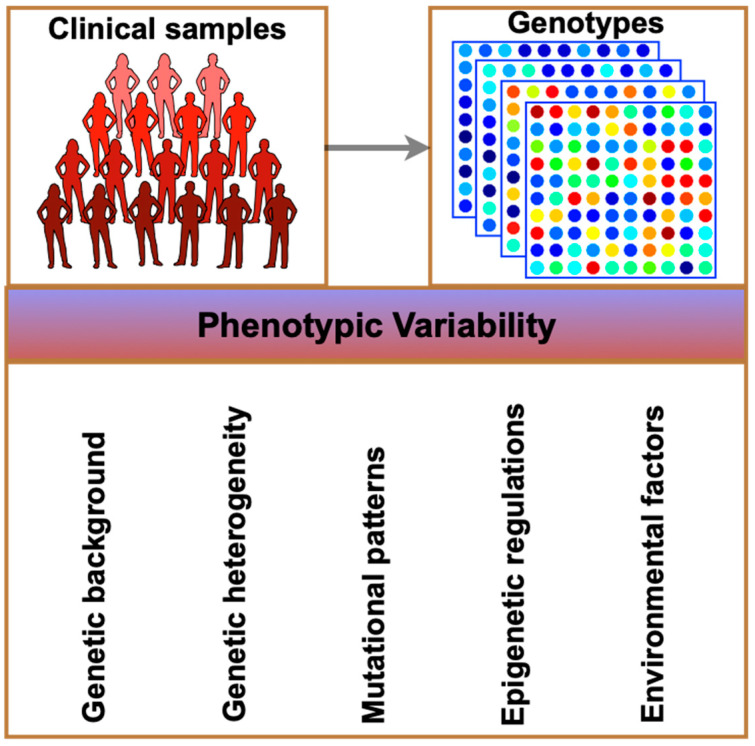
Factors associated with phenotype variation.

**Figure 3 pharmaceuticals-17-00940-f003:**
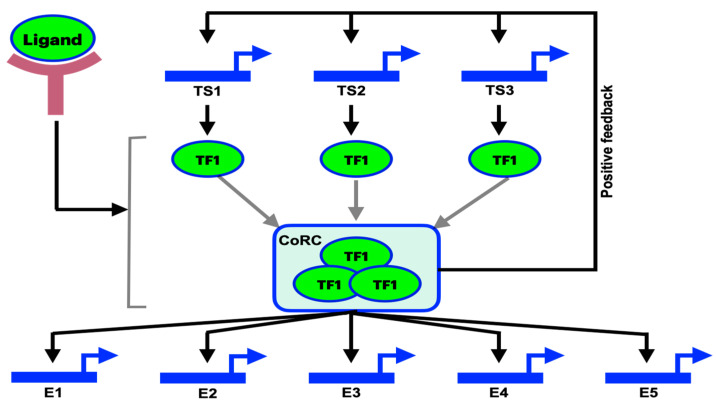
Cell-type identity regulatory signature [1]. A paradigm for identifying the kind of cell. A limited group of terminal selector genes (TS1 to TS3) produce transcription factors (TF1 to TF3), which are altered when ligands connect to them and create a core regulatory complex (CoRC) via activating signaling pathways. The molecular agent known as CoRC is responsible for both maintaining its own expression and controlling the downstream effector genes (E1 to E5). To sum up, the terminal selector transcription factors work together to produce a CoRC, which controls the expression of genes exclusive to a certain cell type and promotes the evolutionary independence of that cell type.

**Figure 4 pharmaceuticals-17-00940-f004:**
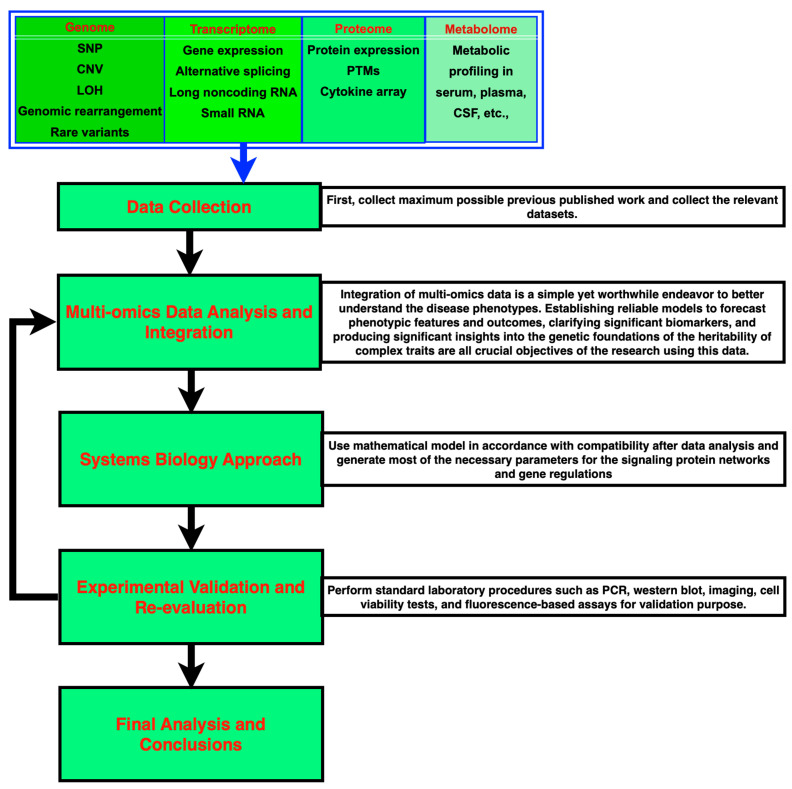
A summary for computational application showing multi-omics data integration and analysis [31].

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
