# Peer review of "Pharmacogenomics: A Genetic Approach to Drug Development and Therapy"

_pharmaceuticals, 2024, doi:10.3390/ph17070940_

Round 1

Reviewer 1 Report

Comments and Suggestions for Authors

Pharmacogenomics: A Genetic Approach to Drug Development and Therapy

The authors give a background about pharmacogenomics and how specific genetics and genotypes can define treatment response.

Also, some factors associated with this such as the environment and patient characteristics have been mentioned in the abstract.

The aims are then formulated, forming a basis for pharmacogenomic applications, techniques and strategies.

Hopefully, these aims will be met.

The intro starts with the role of SNPs and genomics as an integral part of this field. The paradigm has also been mentioned.

Some factors that affect response to medicine are also mentioned and a figure accompanies this.

Figure 1 could be expanded upon a bit more. Some concepts have been given in the last 2 boxes that have not been referred to yet.

The aims then get formulated in line 58. I would have thought that the aims should be the last section of the intro but the intro continues after that.

Genomics and precision medicine might cross paths in NGS. The topic moves to gDNA and somatic mutations and some content is provided about libraries and target capture and some other steps are recounted such as mapping to the reference genome. I would like to make two points here; 1- why go into this detail here and what point are the authors trying to make? 2- There are few references across the study, especially in this section ending with (figure 1), line 73.

Unexpectedly, the topic moves to morbidity and mortality with specific patients following drug treatment, how is this linked to the previous paragraph? Some reasons why patients might respond differently to drugs are mentioned. An earlier section discussed this too so maybe they can be consolidated to give a better logical flow of topics.

Then biomarkers are introduced, followed by epigenetics. 

Overall, the intro does not have an understandable flow of topics. Some topics have been added, each in a paragraph, but I am not sure by what logic.

I think that the intro needs more work and the coherence of it could be improved.

Section 2. As such, the authors have reintroduced pharmacogenomics from scratch. Correlating genotypes with drug response forecast. Again not enough references are made, large section bear no reference at all, so it is hard to tell if these are the personal opinions of the authors or indeed have been derived from another source.

Then drug responsiveness is another time linked to P450, if this has been already discussed, I am not sure if repeating it is helpful.

Also, the authors should try to give specific examples to give some specificity to their overview-type content.

The next paragraph then talks about how many diseases are not monogenic and therefore might be a lot more complex (polygenic). This becomes more complex in that drugs will also have multiple targets and I think this is a good point. The next topic is GWAS which is useful for understanding polygenic diseases. 

The authors refer to Figure 1 multiple times, maybe they could design another figure.

In line 151, the authors then refer to pharmacogenomics applications but then mention some drugs and specific genes or genotypes. What do the authors mean by applications, that could encapsulate some many different categories of things? Do they mean drugs or genotypes that have been studied using pharmacogenomics techniques? I’m guessing this is the case since clinical trials are then mentioned. This paragraph at least gives some specific examples and that is much better (warfarin). No references have been used.

In general, the flow of topics is not great across the manuscript. Perhaps the authors could have started with genomes, genotypes, drugs, linked genotypes to drug response, tools used etc, clinical implications and possible trials, and future directions as something that would give a gradual buildup to pharmacogenomics. They go backwards and forwards between topics that makes understanding the core message a bit difficult.

The topic on HER and EMR does not feel the logical continuation of the paragraph before it that is about specific drugs and genotypes.

Section 3, genetic causes, as mentioned should come up before this. As such this topic here discusses ADME, PK-PD but not a lot is mentioned about specific phenotypes. The topic of P450 is appearing again but perhaps other examples could be given. Response to EGFP and gefitinib is good and the readers would like to see more of this. If the authors remain in an overview style coverage, they would provide enough specific examples and the reader is not going to gain a deeper understanding of the topic.

4. I have mentioned that the topics of this manuscript do not follow a logical order. The future directions are placed before the next sections that focus on epigenetics etc. I would suggest bringing this genetics/epigenetics section forward, then talking about drugs, linkage of genotype and drug response etc and future direction should be the last or one of the last topics. Each section should also have a good number of primary studies that specifically match that topic (similar to the EGF example above), otherwise, not enough depth will be achieved.

Again, with the restructuring of the study, drug response will be placed in a better location. The mention of some drugs that have a pharmacogenomic profile is useful but the authors could expand on one of them and tell the reader in a comprehensive but concise manner the story behind it.

The topics on the range of doses and also combination therapy are useful. The former should have a specific example, the latter does already have one, so that is ok. 

6. the gap of knowledge section could be placed just after the genetics/ epigenetics sections have been introduced. The SERT example is relevant too.

7. This section could be placed before the gap of knowledge. One would first tell what is known and then mention the unknowns. Topics such as alternative splicing and mutations and how they may induce variation are all very important and with the new order of topics they will read much better. Section 8 would be better suited to an earlier section closer to genetics.

Overall, as mentioned, there is some merit to this study, the authors have spent time and energy investigating the topic but can use some help with the order of the topics, and good solid examples that support each relevant topic.

Also, the intro had some method-like content but I didn’t much about it later, perhaps if a short methods section could be added, that would help.

I would also suggest more figures/ schemes are added. Perhaps a table which looks at summarising some of the key aspects may be useful.

Comments on the Quality of English Language

Some editing required

Author Response

Response to Reviewer 1.

The authors give a background about pharmacogenomics and how specific genetics and genotypes can define treatment response.

Also, some factors associated with this such as the environment and patient characteristics have been mentioned in the abstract.

The aims are then formulated, forming a basis for pharmacogenomic applications, techniques and strategies.

Authors: We appreciate that you have spent your precious time.

Hopefully, these aims will be met.

The intro starts with the role of SNPs and genomics as an integral part of this field. The paradigm has also been mentioned.

Some factors that affect response to medicine are also mentioned and a figure accompanies this.

Figure 1 could be expanded upon a bit more. Some concepts have been given in the last 2 boxes that have not been referred to yet.

Authors: We agree and have updated.

The aims then get formulated in line 58. I would have thought that the aims should be the last section of the intro but the intro continues after that.

Authors: We agree and have modified.

Genomics and precision medicine might cross paths in NGS. The topic moves to gDNA and somatic mutations and some content is provided about libraries and target capture and some other steps are recounted such as mapping to the reference genome. I would like to make two points here; 1- why go into this detail here and what point are the authors trying to make? 2- There are few references across the study, especially in this section ending with (figure 1), line 73.

Authors: We understand the concern and we just wanted to give some basics of sample processing. By taking care of your concern, we updated the text.

Unexpectedly, the topic moves to morbidity and mortality with specific patients following drug treatment, how is this linked to the previous paragraph? Some reasons why patients might respond differently to drugs are mentioned. An earlier section discussed this too so maybe they can be consolidated to give a better logical flow of topics.

Then biomarkers are introduced, followed by epigenetics. 

Authors: We agree and have addressed it made the flow better for clear readability.

Overall, the intro does not have an understandable flow of topics. Some topics have been added, each in a paragraph, but I am not sure by what logic. I think that the intro needs more work and the coherence of it could be improved.

Authors: We agree and have edited and modified the introduction section.

Section 2. As such, the authors have reintroduced pharmacogenomics from scratch. Correlating genotypes with drug response forecast. Again not enough references are made, large section bear no reference at all, so it is hard to tell if these are the personal opinions of the authors or indeed have been derived from another source.

Authors: We have updated the text and the references.

Then drug responsiveness is another time linked to P450, if this has been already discussed, I am not sure if repeating it is helpful.

Also, the authors should try to give specific examples to give some specificity to their overview-type content.

Authors: Agreed and reduced the text.

The next paragraph then talks about how many diseases are not monogenic and therefore might be a lot more complex (polygenic). This becomes more complex in that drugs will also have multiple targets and I think this is a good point. The next topic is GWAS which is useful for understanding polygenic diseases. 

Authors: Thanks.

The authors refer to Figure 1 multiple times, maybe they could design another figure.

Authors: Well, we feel the audiences to have better focus on figure 1 but still we updates the text further and included few more new things.

In line 151, the authors then refer to pharmacogenomics applications but then mention some drugs and specific genes or genotypes. What do the authors mean by applications, that could encapsulate some many different categories of things? Do they mean drugs or genotypes that have been studied using pharmacogenomics techniques? I’m guessing this is the case since clinical trials are then mentioned. This paragraph at least gives some specific examples and that is much better (warfarin). No references have been used.

Authors: Agreed and added reference.

In general, the flow of topics is not great across the manuscript. Perhaps the authors could have started with genomes, genotypes, drugs, linked genotypes to drug response, tools used etc, clinical implications and possible trials, and future directions as something that would give a gradual buildup to pharmacogenomics. They go backwards and forwards between topics that makes understanding the core message a bit difficult.

The topic on HER and EMR does not feel the logical continuation of the paragraph before it that is about specific drugs and genotypes.

Authors: We understand the concern and we designed it in the same way as you have suggested and we also understand the source of confusion so we modified the text.

Section 3, genetic causes, as mentioned should come up before this. As such this topic here discusses ADME, PK-PD but not a lot is mentioned about specific phenotypes. The topic of P450 is appearing again but perhaps other examples could be given. Response to EGFP and gefitinib is good and the readers would like to see more of this. If the authors remain in an overview style coverage, they would provide enough specific examples and the reader is not going to gain a deeper understanding of the topic.

Authors: We have addressed the concern.

  1. I have mentioned that the topics of this manuscript do not follow a logical order. The future directions are placed before the next sections that focus on epigenetics etc. I would suggest bringing this genetics/epigenetics section forward, then talking about drugs, linkage of genotype and drug response etc and future direction should be the last or one of the last topics. Each section should also have a good number of primary studies that specifically match that topic (similar to the EGF example above), otherwise, not enough depth will be achieved.

Authors: We have addressed the concern.

 Again, with the restructuring of the study, drug response will be placed in a better location. The mention of some drugs that have a pharmacogenomic profile is useful but the authors could expand on one of them and tell the reader in a comprehensive but concise manner the story behind it. 

The topics on the range of doses and also combination therapy are useful. The former should have a specific example, the latter does already have one, so that is ok. 

Authors: We agree and addressed the concern.

  1. the gap of knowledge section could be placed just after the genetics/ epigenetics sections have been introduced. The SERT example is relevant too.

Authors: We agree and xx.

  1. This section could be placed before the gap of knowledge. One would first tell what is known and then mention the unknowns. Topics such as alternative splicing and mutations and how they may induce variation are all very important and with the new order of topics they will read much better. Section 8 would be better suited to an earlier section closer to genetics.

Authors: We agree and xx.

Overall, as mentioned, there is some merit to this study, the authors have spent time and energy investigating the topic but can use some help with the order of the topics, and good solid examples that support each relevant topic.

Also, the intro had some method-like content but I didn’t much about it later, perhaps if a short methods section could be added, that would help.

Authors: We agree and very short details were added now.

I would also suggest more figures/ schemes are added. Perhaps a table which looks at summarising some of the key aspects may be useful.

Authors: Authors: Overall, we have addressed all the concerns and we could clearly see the improvement in the manuscript quality and we hope to satisfy the reviewers. We also appreciate the efforts of the reviewers.

Reviewer 2 Report

Comments and Suggestions for Authors

The review is written on a current topic affecting pharmacogenomics and related opportunities for personalized medicine.

However, the information presented in it is not sufficiently generalized. In its current form, the authors cite only 55 literary sources (13 pages of manuscript, without a list of references), which does not allow them to fully cover the problem. In this regard, there are large areas of text in which there are no citations at all. I think it would be more correct to increase the number of citations to at least 100, while focusing on works published over the past 5 years.

It is also important to accumulate knowledge in at least one table. There are 11 chapters in the current review, which provide important insight into the problem, but the data in them is scattered, there is not a single table that would benefit readers, for example, in the form of a list of genetic markers discovered in the last 5 years, or a list of recent works , aimed at one or another branch of pharmacogenomics. Not counting the Introduction and Conclusion, the review contains 9 chapters, illustrated with only 4 figures. Based on this, in some Sections there are repeated references to the Figure from the Previous Section. It would be logical to increase the number of Figures in the review in order to facilitate the perception of the material for readers.

In its current form, the review contains 4 Figures, of which Figure 3 is completely borrowed from the article DOI: 10.1038/nrg.2016.127, published in Nat Rev Genet (2016), however, the authors of the current review for some reason changed the original color scheme, in addition to the current layout The lower part of the Figure is cut off, which deprives it of any semantic content.

In addition, the authors significantly shortened the original caption to Figure 3 to “Cell-type identity regulatory signature.” In its original form, the signature was more meaningful: “A model of cell type identity determination. A small set of terminal selector genes (TS1 to TS3) are producing transcription factors (TF1 to TF3), which are modified through the activation of signalling pathways upon binding of a ligand (green oval) and form a core regulatory complex (CoRC). The CoRC is the molecular agent that regulates the downstream effector genes (E1 to E5) and maintains its own expression. In summary, the terminal selector transcription factors cooperatively interact and form a CoRC to regulate cell type-specific gene expression and to enable cell type evolutionary independence. Grey arrows represent translation and complex formation. Black arrows represent regulation." In addition, only a part of the original Figure is shown (its Figure 2A part), and Figures 2Ba, 2Bb, which reveal the molecular mechanisms of interaction in more detail, are not shown.

Based on this, the question arises whether it is really necessary to cite a Figure from another review, especially since all the other 3 Figure are original. Maybe it’s enough to provide a link to this review (DOI: 10.1038/nrg.2016.127) without providing the Figure? Or at least give the drawing uncropped (from below). In addition, the meaning of the drawing must be fully disclosed through its signature, and understandable outside the context of the article. Therefore, return the caption to Figure 3, as in the original article, where the abbreviations and symbols in the rusun are deciphered, as well as the interactions that occur.

Based on the above, also significantly expand the captions to Figures 1 and 2, describing in detail the processes depicted. Also expand the caption to Figure 4. In its current form (A summary for computational application) it looks too general, it may refer to many other Rusunki describing computational application.

Comments

Lines 30-31

“Themes from theory or experimentation may appear in this special issue.” (c)

This phrase is somewhat unclear within the framework of this review, since it is not clear what theoretical or experimental work may appear in this special issue. Readers should additionally know that one of the review co-authors, Dr. Mohammad Mobashir is also the guest editor of the special issue in which this review is submitted. However, this special issue is closed (March 31, 2024), so it is not clear why the submitted review talks about publications of the special issue in the future tense. I recommend removing this phrase altogether, since it is of an advertising nature and does not address the essence of the problem.

Line 42

The citation begins with the second literary source; it is not clear where literary source 1 “disappeared”.

Figure 1.

In the title of the Figure you talk about “Integration of multi-omics data”, however, the figure itself shows that the Clinical samples collection (1st part of the figure) are subject to only genotyping (2nd part of the figure), on the basis of which further conclusions are given. Genotyping involves only genome analysis (genomics). You are not talking about other omics approaches (such as metabolomics, transcriptomics, lipidomics and others, which can also be used by analyzing the corresponding Clinical samples collection). In this case, you need to remove the word multi-omics data, or expand the analysis of samples, where the current version of the Figure only talks about genotyping (genomics).

Lines 105-119

This paragraph is not illustrated by any citations. Please provide examples of scientific publications so that the theses you put forward would be more specific.

Author Response

Response to Reviewer 2.

The review is written on a current topic affecting pharmacogenomics and related opportunities for personalized medicine.

However, the information presented in it is not sufficiently generalized. In its current form, the authors cite only 55 literary sources (13 pages of manuscript, without a list of references), which does not allow them to fully cover the problem. In this regard, there are large areas of text in which there are no citations at all. I think it would be more correct to increase the number of citations to at least 100, while focusing on works published over the past 5 years.

Authors: We agree and have increased the references further.

It is also important to accumulate knowledge in at least one table. There are 11 chapters in the current review, which provide important insight into the problem, but the data in them is scattered, there is not a single table that would benefit readers, for example, in the form of a list of genetic markers discovered in the last 5 years, or a list of recent works , aimed at one or another branch of pharmacogenomics. Not counting the Introduction and Conclusion, the review contains 9 chapters, illustrated with only 4 figures. Based on this, in some Sections there are repeated references to the Figure from the Previous Section. It would be logical to increase the number of Figures in the review in order to facilitate the perception of the material for readers.

Authors: We understand and that is the reason we have also provided one figure which could help the audiences to learn and implement the approach for similar goals.

In its current form, the review contains 4 Figures, of which Figure 3 is completely borrowed from the article DOI: 10.1038/nrg.2016.127, published in Nat Rev Genet (2016), however, the authors of the current review for some reason changed the original color scheme, in addition to the current layout The lower part of the Figure is cut off, which deprives it of any semantic content.

Authors: Yes, that’s right and I have clearly mentioned that I have redrawn this figure and thus adapted from the given reference and here our goal was to display a simple model for cell type identification which is of potential significance for targeted therapeutics.

In addition, the authors significantly shortened the original caption to Figure 3 to “Cell-type identity regulatory signature.” In its original form, the signature was more meaningful: “A model of cell type identity determination. A small set of terminal selector genes (TS1 to TS3) are producing transcription factors (TF1 to TF3), which are modified through the activation of signalling pathways upon binding of a ligand (green oval) and form a core regulatory complex (CoRC). The CoRC is the molecular agent that regulates the downstream effector genes (E1 to E5) and maintains its own expression. In summary, the terminal selector transcription factors cooperatively interact and form a CoRC to regulate cell type-specific gene expression and to enable cell type evolutionary independence. Grey arrows represent translation and complex formation. Black arrows represent regulation." In addition, only a part of the original Figure is shown (its Figure 2A part), and Figures 2Ba, 2Bb, which reveal the molecular mechanisms of interaction in more detail, are not shown.

Based on this, the question arises whether it is really necessary to cite a Figure from another review, especially since all the other 3 Figure are original. Maybe it’s enough to provide a link to this review (DOI: 10.1038/nrg.2016.127) without providing the Figure? Or at least give the drawing uncropped (from below). In addition, the meaning of the drawing must be fully disclosed through its signature, and understandable outside the context of the article. Therefore, return the caption to Figure 3, as in the original article, where the abbreviations and symbols in the rusun are deciphered, as well as the interactions that occur.

Authors: We agree and updated the section for further details.

Based on the above, also significantly expand the captions to Figures 1 and 2, describing in detail the processes depicted. Also expand the caption to Figure 4. In its current form (A summary for computational application) it looks too general, it may refer to many other Rusunki describing computational application.

Authors: We addressed almost all the concerns and have added two more tables also.

Comments

Lines 30-31

“Themes from theory or experimentation may appear in this special issue.” (c)

This phrase is somewhat unclear within the framework of this review, since it is not clear what theoretical or experimental work may appear in this special issue. Readers should additionally know that one of the review co-authors, Dr. Mohammad Mobashir is also the guest editor of the special issue in which this review is submitted. However, this special issue is closed (March 31, 2024), so it is not clear why the submitted review talks about publications of the special issue in the future tense. I recommend removing this phrase altogether, since it is of an advertising nature and does not address the essence of the problem.

Line 42

The citation begins with the second literary source; it is not clear where literary source 1 “disappeared”.

Authors: We have addressed the mentioned comments and we feel the there is a lot improvements now. 

Figure 1.

In the title of the Figure you talk about “Integration of multi-omics data”, however, the figure itself shows that the Clinical samples collection (1st part of the figure) are subject to only genotyping (2nd part of the figure), on the basis of which further conclusions are given. Genotyping involves only genome analysis (genomics). You are not talking about other omics approaches (such as metabolomics, transcriptomics, lipidomics and others, which can also be used by analyzing the corresponding Clinical samples collection). In this case, you need to remove the word multi-omics data, or expand the analysis of samples, where the current version of the Figure only talks about genotyping (genomics).

Authors: Well this is how in general the people think about pharmacogenomics that is why we have drawn a such layout in figure 1. 

This paragraph is not illustrated by any citations. Please provide examples of scientific publications so that the theses you put forward would be more specific.

Authors: Overall, we have addressed all the concerns and we could clearly see the improvement in the manuscript quality and we hope to satisfy the reviewers. We also appreciate the efforts of the reviewers.

Round 2

Reviewer 2 Report

Comments and Suggestions for Authors

The authors generally took into account the necessary recommendations, approximately doubled the number of citations, and accumulated knowledge in two new tables. The material has been significantly expanded, making the review more holistic and meaningful.

1) However, the citation itself in the Tables is used poorly. All citations refer to the title of the table and not to its individual rows. For example:

Table 1. Essential fundamental methods [87] for pharmacogenetics and genomics genotype analysis[2-4, 17, 48, 55, 58, 63, 87, 88]. Table 2. Bioinformatics databases and software tools [87] for pharmacogenetics and genomics[76][2-5, 9, 17, 46, 48, 49, 55, 58, 63, 88, 706 89].

It is not customary to cite in tables in this form, since it makes it difficult for the reader to compare information with one or another source of citation. It is customary to provide relevant citations on a separate line. In this form, the authors presented the review as such, often all citations are given at the end of the paragraph, rather than being correlated with individual sentences. For the text of a review, such a citation representation can still be allowed (although the reader will have to figure out which quotation corresponds to which sentence), but for a table such a citation representation is unacceptable. Perhaps the authors did not have enough time to arrange the citations in the appropriate rows of the table. In this case, they need to spend more time refining the manuscript so that it is easier for readers to perceive the material presented.

2) In addition, the tables are not integrated into the review text. There are no links to the tables, which needs to be done. The tables are placed in tandem behind Figure 4, looking like the Appendix. Add connective text that integrates your tables into the review text.

3) The authors also write: “Authors: We understand and that is the reason we have also provided one figure which could help the audiences to learn and implement the approach for similar goals.” However, in the resubmitted version there are no new Figures, there are 4 of them, as in the original version of the review.

4) The authors did not respond to comment

“Line 42

The citation begins with the second literary source; it is not clear where literary source 1 “disappeared.”!”

In the current version, as before, the introduction begins with the phrase: “Pharcogenetic research over a long period of time has demonstrated how genetic 36 variants affect drug response in a broad way [2-5]. ”

Where is the link to source 1?

5) Also, the authors did not answer the question

“Figure 1.

In the title of the Figure you talk about “Integration of multi-omics data”, however, the figure itself shows that the Clinical samples collection (1st part of the figure) are subject to only genotyping (2nd part of the figure), on the basis of which further conclusions are given. Genotyping involves only genome analysis (genomics). You are not talking about other omics approaches (such as metabolomics, transcriptomics, lipidomics and others, which can also be used by analyzing the corresponding Clinical samples collection). In this case, you need to remove the word multi-omics data, or expand the analysis of samples, where the current version of the Figure only talks about genotyping (genomics).”

The caption to Figure 1 talks about multi-omics data (approach where the data sets of different omic groups are combined during analysis), while the figure talks only about genomics.

Author Response

The authors generally took into account the necessary recommendations, approximately doubled the number of citations, and accumulated knowledge in two new tables. The material has been significantly expanded, making the review more holistic and meaningful.

1) However, the citation itself in the Tables is used poorly. All citations refer to the title of the table and not to its individual rows. For example:

Table 1. Essential fundamental methods [87] for pharmacogenetics and genomics genotype analysis[2-4, 17, 48, 55, 58, 63, 87, 88]. Table 2. Bioinformatics databases and software tools [87] for pharmacogenetics and genomics[76][2-5, 9, 17, 46, 48, 49, 55, 58, 63, 88, 706 89].

It is not customary to cite in tables in this form, since it makes it difficult for the reader to compare information with one or another source of citation. It is customary to provide relevant citations on a separate line. In this form, the authors presented the review as such, often all citations are given at the end of the paragraph, rather than being correlated with individual sentences. For the text of a review, such a citation representation can still be allowed (although the reader will have to figure out which quotation corresponds to which sentence), but for a table such a citation representation is unacceptable. Perhaps the authors did not have enough time to arrange the citations in the appropriate rows of the table. In this case, they need to spend more time refining the manuscript so that it is easier for readers to perceive the material presented.

Response: We agree and have added the relevant references in relevant places.

2) In addition, the tables are not integrated into the review text. There are no links to the tables, which needs to be done. The tables are placed in tandem behind Figure 4, looking like the Appendix. Add connective text that integrates your tables into the review text.

Response: We partially agree and still updated the text just to satisfy this comment.

3) The authors also write: “Authors: We understand and that is the reason we have also provided one figure which could help the audiences to learn and implement the approach for similar goals.” However, in the resubmitted version there are no new Figures, there are 4 of them, as in the original version of the review.

Response: We agree and have addressed it.

4) The authors did not respond to comment

“Line 42

The citation begins with the second literary source; it is not clear where literary source 1 “disappeared.”!”

Response: To address this concern first thing is that we uploaded the pdf file and to insert citation endnote has been used.

In the current version, as before, the introduction begins with the phrase: “Pharcogenetic research over a long period of time has demonstrated how genetic 36 variants affect drug response in a broad way [2-5]. ”

Where is the link to source 1?

 Response: Here, we have not mentioned at all the number that is 36 variants. We just bring the generalized concept that how genetic variants affect drug response for which we have mentioned the references.

5) Also, the authors did not answer the question

“Figure 1.

In the title of the Figure you talk about “Integration of multi-omics data”, however, the figure itself shows that the Clinical samples collection (1st part of the figure) are subject to only genotyping (2nd part of the figure), on the basis of which further conclusions are given. Genotyping involves only genome analysis (genomics). You are not talking about other omics approaches (such as metabolomics, transcriptomics, lipidomics and others, which can also be used by analyzing the corresponding Clinical samples collection). In this case, you need to remove the word multi-omics data, or expand the analysis of samples, where the current version of the Figure only talks about genotyping (genomics).”

The caption to Figure 1 talks about multi-omics data (approach where the data sets of different omic groups are combined during analysis), while the figure talks only about genomics.

Response: We partially agree and have modified figure 4 to address this concern and also updated the text for properly linking the text. Furthermore, we have included one of the highly relevant reference that ref nr 31 (Ritchie M.D. Nature Reviews Genetics volume 16, pages85–97 (2015)). Overall, we have addressed all the concerns and we could clearly see the improvement in the manuscript quality and we hope to satisfy the reviewers. We also appreciate the efforts of the reviewers.
